Effects of education methods on self-efficacy of smoking cessation counseling among medical students

Cho Ara 1
Lee Jeonggyu 1
Kim YunJin 1
Cho Byung Mann 2
Lee Sang Yeoup saylee@pnu.edu 3 4
Kong Eunhee 5
Kim Minjeong 6
Kim Jinseung 7
Jung Dong Sik 8
Han Seongho 9
1 Department of Family Medicine, Pusan National University School of Medicine , Yangsan , South Korea
2 Department of Preventive Medicine, Pusan National University School of Medicine , Yangsan , South Korea
3 Department of Medical Education, Pusan National University School of Medicine , Yangsan , South Korea
4 Department of Family Medicine and Research Institute of Convergence of Biomedical Science and Technology, Pusan National University Yangsan Hospital , Yangsan , South Korea
5 Department of Family Medicine, Kosin University College of Medicine , Busan , South Korea
6 Department of Medical Education, Kosin University College of Medicine , Busan , South Korea
7 Department of Family Medicine, Inje University College of Medicine , Busan , South Korea
8 Department of Infective Medicine, Dong-A University College of Medicine , Busan , South Korea
9 Department of Family Medicine, Dong-A University College of Medicine , Busan , South Korea
Sewall Julia
Electronic publication date: 2021 May 7
Publication date: 2021
Volume: 9
Electronic Location ID: e11408
Received 2021 Jan 28; Accepted 2021 Apr 14
Copyright: ©2021 Cho et al.
Copyright year: 2021
Copyright holder: Cho et al.
License: This is an open access article distributed under the terms of the Creative Commons Attribution License, which permits unrestricted use, distribution, reproduction and adaptation in any medium and for any purpose provided that it is properly attributed. For attribution, the original author(s), title, publication source (PeerJ) and either DOI or URL of the article must be cited.
License URL: https://creativecommons.org/licenses/by/4.0/

Keywords: Counseling, Education methods, Medical students, Self-efficacy, Smoking cessation

Funding: The authors received no funding for this work.

==============================
Background

Medical students need to receive training in providing smoking cessation counseling to provide effective smoking cessation interventions to smokers when they become doctors. This study examined the smoking cessation education curricula and factors affecting counseling self-efficacy (CSE) in smoking cessation treatment among medical students.

Methods

In a multicenter cross-sectional study, we obtained demographic information, personal history of tobacco use and intention to quit smoking, exposure to secondhand smoke in the school premises during the past week, the experience of learning about tobacco in each medical school, tobacco-related medical knowledge, and self-efficacy in smoking cessation counseling on medical students of four Korean medical schools.

Results

Among 1,416 medical students eligible, 313 (22.1%) students completed a self-administered questionnaire. Only 20.3% of the students reported positive CSE on smoking cessation. The factors affecting positive CSE were scores of ≥ 60 on tobacco-related medical knowledge, smoking experience, and blended learning (p = 0.014, 0.005, and 0.015, respectively).

Conclusion

This study shows that high scores in tobacco-related medical knowledge and blended learning are correlated with positive CSE for smoking cessation counseling.

Introduction

Tobacco smoking still ranks among the most common causes of preventable deaths and morbidity world-wide that medical students will encounter in their future lives as physicians (GBD 2015 Risk Factors Collaborators, 2016). According to a Cochrane analysis, physician counselling is among the most cost-effective clinical interventions for smoking cessation. It plays an important role in facilitating smokers’ attempts to quit, and consequently, in lowering smoking rates (Stead et al., 2013; Zwar, Mendelsohn & Richmond, 2014). The US Preventive Services Task Force recommends that based on patients’ willingness to quit smoking, physicians should provide smoking cessation counselling at each visit to every patient who uses tobacco (US Preventive Services Task Force, 2009; Fiore & Baker, 2011). Providing adequate education is a priority for encouraging doctors to set up smoking cessation counseling practices and in raising their confidence in tobacco cessation interventions (Victor et al., 2010). However, after becoming a doctor there have been reports of poor participation in practical training education on tobacco cessation counseling (Caplan, Stout & Blumenthal, 2011; Champassak et al., 2014; Herold et al., 2016). It is therefore necessary for medical schools to provide proper smoking cessation education to medical students, the doctors of the future, but such education has been found to be insufficient according to surveys conducted by medical schools in most countries such as Canada (Loranger, Simms & Pipe, 2018), the United States (Warren et al., 2011), the United Kingdom (Raupach et al., 2015), Germany (Strobel et al., 2012), and Italy (Grassi et al., 2012).

It was pointed out that the traditional lecture-based method was insufficient for training medical professionals in providing counseling and supporting successful smoking cessation (Park, Park & Hwang, 2019). Recently, as an alternative, various new educational methods were proposed to improve the self-efficacy of medical students as counselors (Stolz et al., 2012; Herold et al., 2016; Ockene et al., 2016). Previous studies have found that the self-assessment score for smoking cessation counseling as well as the objective structured clinical examination score was higher in medical students who received smoking cessation training through role playing and interaction with real patient methods (Stolz et al., 2012) or a multimodal and interactive teaching module on smoking cessation, including online learning material, seminars, and practical skills training than students who only received lectures (Herold et al., 2016). Self-efficacy is more than just a simple belief in one’s ability to succeed in specific situations or accomplish tasks. It plays an important role in effective patient care by lowering physicians’ anxiety about their behavior’s success or failure. It is also a useful indicator that improves physicians’ performance and patient outcomes by enabling them to actually perform behaviors or attain specific outcomes successfully (Grassi et al., 2012; Schiele et al., 2014). To the best of our knowledge, no research has been conducted in Korea on how much smoking cessation education was being implemented, its effectiveness, and the medical students’ degree of self-efficacy in smoking cessation treatment. In this regard, the aim of this study was to scrutinize the smoking cessation education curricula, students’ smoking behaviors, medical knowledge related to tobacco, and counseling self-efficacy (CSE) in smoking cessation counseling in Korean medical schools. We also analyzed factors affecting their CSE in smoking cessation treatment.

Methodology

Study design and subjects

This multi-center, cross-sectional survey was conducted in four medical schools (A–D) located in Busan, an urban city of South Korea, by administering an online questionnaire to 1,416 students between December 2016 to January 2017. Korea’s medical school degree course currently consists of the first two years of the pre-medical phase and then the pre-clerkship or pre-clinical phase (Year 1 & 2) and the clerkship or clinical phase (Year 3 & 4). A week prior to the survey, notices of the survey were posted, and text messages with a link to the online survey were sent to the students, who were given the option of completing the entire questionnaire via internet. At the beginning of the online survey, we provided the following information: participation is voluntary, the responses will be kept confidential and anonymous, students must not fear that their non-participation in this study may jeopardize their grade or progress in any way. Electronic informed consent was obtained by submitting the completed questionnaire from each participant prior to commencement of the questionnaire. A total of 313 students responded to the questionnaire. In addition, coordinators from each medical school were asked to provide information on smoking cessation education (e.g., the existence of a tobacco curriculum, during which year it was taught, tobacco content in the curriculum, and format of teaching methods). Only objective data were received from each school’s coordinator, not personal opinions that could cause potential bias. In Korea, there are a total of 40 medical schools nationwide.

Questionnaire

A self-administered questionnaire was developed by tobacco treatment specialists and supervised by a medical education expert. The 23 questions of the questionnaire were divided into the following six parts; (1) questions 1–4: demographics such as gender, age, medical school, and grade; (2) questions 5–10: details of personal history such as smoking status, age at smoking initiation, types of tobacco used, smoking duration and intensity, duration of smoking cessation, and intention to quit smoking; (3) question 11: exposure to secondhand smoke (SHS) at school in the past week; (4) questions 12 and 13: the experience of learning in medical schools about tobacco-related medical knowledge and smoking cessation counseling techniques; (5) questions 14–22: tobacco-related medical knowledge such as smoking prevalence among adults in Korea, tobacco-related mortality, tobacco-related illnesses, the effect of SHS on stroke, the effect of brief smoking cessation interventions, smoking cessation counseling techniques, nicotine dependence treatment, length of time after quitting when the risk of cardiovascular diseases is reduced or returns to normal, and the benefits of smoking cessation in reducing sudden and premature deaths; and (6) question 23: CSE in smoking cessation (See Supplemental information).

Definitions of terms used in the present study

Non-smokers were defined as those who had no smoking experience or had smoked less than 100 cigarettes in their lifetime. Current smokers who had smoked 100 or more cigarettes in their lifetime were classified as daily smokers if they had smoked any tobacco product at least once a day, occasional smokers if they did not smoke daily, and former smokers if they were currently not smoking (Hall et al., 2019). The four medical schools integrated different content during different schoolyears within the curriculum. However, based on its teaching methods, smoking cessation education was classified into lecture-based (A and C schools) and blended education (B and D schools). In the former case, smoking cessation education was provided only in the form of lectures, while in the latter, in addition to lectures, one or more interactive teaching methods such as discussions, tobacco-focused standardized patients simulation, or role plays were implemented. CSE in the treatment of smoking cessation was assessed on a 5-point Likert scale, with scores of 4 or higher and 3 or less being interpreted as high and low self-efficacy, respectively (Triantafylidis et al., 2019).

How the tobacco-related knowledge items were scored

The correct answers to smoking prevalence among adults in Korea and tobacco-related mortality were based on statistics published by the Korean Centers for Disease Control released closest to the time of the study. For questions smoking prevalence among adults in Korea, tobacco-related mortality, and length of time after quitting when the risk of cardiovascular diseases is reduced or returns to normal which required numerical value answers, if the answer was within 10% of the reference value it was considered as correct. For the nine questions 14–22 on medical knowledge related to smoking, six questions with one correct answer were given 2 points each, while the remaining three questions (tobacco-related illnesses, smoking cessation counseling techniques, and nicotine dependence treatment) with multiple correct answers were given 1 point for each correct answer. Question 19 (smoking cessation counseling techniques) included the order of correct answers in the scoring standard. Questions 18 (the effect of brief smoking cessation interventions), 19 (smoking cessation counseling techniques), and 22 (the benefits of smoking cessation in reducing sudden and premature deaths) were answered using a 5-point Likert scale and a score of 4 or more was considered correct. The total score was 23 points.

Statistical analysis

We used the calculator provided from Raosoft® to determine the sample size representing the study population. The calculated sample size was 303, assuming a response distribution of 50:50 on the survey (population size 1,416), with 5% margin of error and 95% confidence level. Data were presented as the mean ± standard deviation, median (range), or frequency (percent). Normality of continuous variables was assessed by the Shapiro–Wilk test. Comparison between groups was done with the two-sample t-test or Mann–Whitney U test, while the chi-squared test or Fisher’s exact test were used as appropriate for continuous and categorical variables, respectively. Multivariable logistic regression with a backward stepwise procedure was applied to select significant independent predictors of high self-efficacy. P values of < .05 were deemed statistically significant. All analyses were performed using IBM SPSS statistics software program version 22.0 (IBM Corp., Armonk, NY).

Ethical consideration

The study was approved by the Institutional Review Board of Pusan National University Yangsan Hospital (IRB No. 05-2016-105) and was performed in accordance with the principles of the Declaration of Helsinki. All subjects gave their informed consent before they participated in the study.

Results

Tobacco control curricular content

The curriculum for smoking cessation education at the four medical schools—A, B, C, and D are presented in Table 1. Smoking cessation education was offered by A and C schools only in the form of lectures; B school provided lectures and role plays for 2 h in 3rd year, and only lectures for 2 h in 4th year. School D provided lectures for one hour in 1st year, and for the next 4 h, allowed the students to practice smoking cessation counseling skills with standardized patients.

Table 1 Smoking cessation education in four medical schools (N  = 313).

School of medicine	A (n = 51)	B (n = 47)	C (n = 149)	D (n = 66)	
Curricula	Hours	Teaching methods	Hours	Teaching methods	Hours	Teaching methods	Hours	Teaching methods	
Year 1 (n = 117)							5	Lecture, SP simulation	
Year 2 (n = 55)	1	Lecture			1	Lecture			
Year 3 (n = 72)			2	Lecture, role plays	3	Lecture			
Year 4 (n = 69)			2	Lecture					
Response rate (%)	16.9	23.2	29.9	16.0	
Notes.

SP, standardized patients.

Response rate and demographic characteristics

Of the 1,416 eligible students from the four medical schools, 313 students (22.1%) completed the online survey. No students partially responded to the survey. The response rate of each school varied from 16.0% to 29.9% (Table 1). However, there is no difference the basic characteristics of age, gender distribution, and year of school (curricular phase) et al. between the respondent group (N = 313) and the population (N = 1,416). The students’ average age was 25.1 ± 2.8 years (range: 20–36 years), and the proportion of females was 34.5%. In 1st, 2nd, 3rd, 4th year, the number of participants were 117 (37.4%), 55 (17.6%), 72 (23.0%), and 69 (22.0%), respectively. When divided according to the method of education, those in the lecture-based learning group were older (p < 0.001), and when divided as pre-clerkship and clerkship phases, the rate of implementing smoking cessation education in the pre-clerkship phase was high (p = 0.004), as shown in Table 2.

Table 2 Demographic characteristics and distribution of correct answers to tobacco- related medical knowledge, total number of correct answers, and positive self-efficacy for smoking cessation according to respondents’ education methods.

Variables	Total (N = 313)	Lecture-based learning group (n = 200)	Blended learning group (n = 113)	p value*	
Female (%)	108 (34.5)	66 (33.0)	42 (37.2)	0.456	
Curricular phase				0.004	
  Pre-clerkship	172 (55.0)	122 (61.0)	50 (44.2)		
  Clerkship	141 (45.0)	78 (39.0)	63 (55.8)		
Smoking status				0.483	
  Non-smoker	251 (80.2)	151 (79.0)	93 (82.3)		
  Smoker	62 (19.8)	42 (21.0)	20 (17.7)	0.431	
   Former smoker	18 (29.0)	12 (28.6)	8 (40.0)		
   Current smoker	44 (71.0)	30 (71.4)	12 (60.0)		
Questions**					
  11. In the last 7 days, have you ever been exposed to    secondhand smoke?	264 (84.3)	165 (82.5)	98 (86.7)	0.355	
  12. Have you ever learned about the harm of smoking at    a medical school?	301 (96.2)	192 (96.0)	109 (96.5)	1.000	
  13. Have you ever had a formal training in smoking    cessation counseling techniques at medical schools?	229 (73.2)	134 (67.0)	95 (84.1)	0.001	
  14. Smoking rate of Korean adults	46 (14.7)	25 (12.5)	21 (18.6)	0.144	
  15. Tobacco-related mortality	7 (2.2)	3 (1.5)	4 (3.5)	0.258	
  16. Tobacco-related illnesses	177 (56.5)	111 (55.5)	66 (58.4)	0.618	
  17. The effect of secondhand smoke on stroke	280 (89.5)	175(87.5)	105 (92.9)	0.134	
  18. The effect of brief smoking cessation interventions	149 (47.6)	74 (37.0)	75 (66.4)	<0.001	
  19. smoking cessation counseling techniques (5As: ask,    advise, assess, assist, arrange follow up)	20 (6.4)	1 (0.5)	19 (16.8)	<0.001	
  20. Nicotine dependence treatment	71 (22.7)	30 (15.0)	41 (36.3)	<0.001	
  21. The length of time when the risk of cardiovascular    diseases is reduced or returns to normal after    quitting	21 (6.7)	12 (6.0)	9 (8.0)	0.505	
  22. The benefits of smoking cessation in reducing    sudden and premature deaths	233 (74.4)	139 (69.5)	94 (83.2)	0.008	
Total number of correct answers	3 (0–8)	3 (0–6)	4 (0–8)	<0.001	
CSE in smoking cessation counseling		(n = 177)	(n = 113)		
   Positive***	64 (20.4)	31 (15.5)	33 (29.2)	0.007	
Notes.

Results are expressed as frequencies (percentages).

* Mann-Whitney test, Chi-Square test or Fishers Exact test.

** Number of students who answered yes to questions 11–13, number of students who answered correctly to .

*** Scores of 4 or higher being interpreted as high self-efficacy. CSE, counseling self-efficacy.

Smoking status and exposure to SHS

As shown in Table 2, 19.8% of the students were smokers, and only two (3.2%) were females. Among the smokers, 83.3% had smoked their first cigarette before entering medical school, 80% had used cigarettes, and nine of them had additional experience of using electronic cigarettes, smokeless tobacco, or rolling tobacco. Eighteen students were former smokers, and the remaining (n = 44) were current smokers (5 were occasional smokers and 39 were daily smokers). Of the current smokers, five smoked more than 20 cigarettes a day. Around 75% of the current smokers revealed that they had tried to quit smoking in the past. As regards quitting smoking, six reported that they were ready to try to quit within the next month, and ten within the next 6 months. Around 84.3% students reported experiencing SHS in the school premises during the past week.

Education experience of tobacco

While 96.2% students reported having learned about tobacco-related knowledge through their curriculum, 73.2% answered that they had received a formal training in smoking cessation counseling techniques that could be applied to patients, with significant differences between groups classified according to the teaching methods (67% of lecture-based learning vs. 84.1% of blended learning, p = 0.001, Table 2).

Tobacco-related medical knowledge

Table 2 shows the students’ correct answer rate for the nine questions on tobacco-related medical knowledge according to the teaching methods. The number of correct answers for the effect of brief smoking cessation interventions, smoking cessation counseling techniques, nicotine dependence treatment, the benefits of smoking cessation in reducing sudden and premature deaths and the whole were statistically significant according to the teaching method (p < 0.001, <0.001, <0.001, = 0.008, and <0.001, respectively). Only 14.7% of all students were aware and had accurate information on the smoking rate of Korean adults, whereas 64.5% had overestimated the smoking prevalence. In contrast, students who underestimated tobacco-related mortality and the length of time when the risk of cardiovascular diseases is reduced or returns to normal after quitting were 37.1% and 52.1%, respectively. In the question about smoking cessation counseling skills, 11.5% students received partial scores due to incomplete answers or mistakes in the order of writing answers, and 82.1% wrote completely incorrect answers or none. On questions relating to medication and nicotine dependence, 42.8% students received partial scores. Table 3 shows tobacco-related medical knowledge scores according to teaching methods based on students’ gender (male, female), grades (pre-clerkship phase, clerkship phase), smoking status (smoker, non-smoker), exposure to SHS, learning of tobacco-related medical knowledge, learning of smoking cessation counseling techniques, and self-efficacy (high, low). Overall, as compared to the lecture-based learning group, the blended learning group’s tobacco-related medical knowledge scores were higher in the following: male and female students, those in the pre-clerkship and clerkship phases, those who had experienced SHS exposure, those who perceived having learnt tobacco-related medical knowledge and smoking cessation counseling techniques, and those having high and low self-efficacy. When converted to 100 points, 39 students (12.5%) scored 60 or higher, and 14 of whom were fourth graders (Table 4).

Table 3 Tobacco-related medical knowledge scores between respondents’ sex, grade, smoking status, responses to , and counseling self-efficacy in smoking cessation counseling according to teaching methods.

Variables	Lecture-based learning group (n = 200)	Blended learning group (n = 113)	p value*	
Total (N = 313)	8.0 (0.0–17.0)	11.0 (0.0–21.0)	<0.001	
Sex				
  Male (n = 205)	8.0 (0.0–17.0)	11.0 (2.0–21.0)	<0.001	
  Female (n = 108)	7.0 (0.0–15.0)	11.0 (0.0–21.0)	<0.001	
Curricular phase				
  Pre-clerkship (n = 172)	7.0 (0.0–15.0)	11.0 (5.0–19.0)	<0.001	
  Clerkship (n = 141)	9.0 (0.0–17.0)	11.0 (0.0–21.0)	0.004	
Smoking status				
  Non-smoker (n = 251)	8.0 (0.0–17.0)	11.0 (0.0–21.0)	<0.001	
  Smoker (n = 62)	9.0 (0.0–13.0)	11.0 (5.0–19.0)	<0.001	
Questions				
  11. In the last 7 days, have you ever been exposed to    secondhand smoke? Yes (n = 264)	8.0 (0.0–17.0)	11.0 (0.0–21.0)	<0.001	
  12. Have you ever learned about the harm of smoking at    a medical school? Yes (n = 301)	8.0 (0.0–17.0)	11.0 (0.0–21.0)	<0.001	
  13. Have you ever had a formal training in smoking    cessation counseling techniques at medical schools?   Yes (n = 229)	8.0 (0.0–17.0)	11.0 (2.0–21.0)	<0.001	
CSE in smoking cessation counseling				
  High** (n = 64)	9.0 (2.0–15.0)	11.5 (2.0–21.0)	0.001	
  Low (n = 249)	8.0 (2.0–17.0)	11.0 (0.0–21.0)	<0.001	
Notes.

Results are expressed as median (range).

* Mann-Whitney test.

** Scores of 4 or higher being interpreted as high self-efficacy. CSE, counseling self-efficacy.

Table 4 Proportion of positive CSE in smoking cessation counseling by sex, curricular phase, smoking status, teaching methods, and scores of ≥60 on tobacco-related medical knowledge.

Variables	Total (N = 313)	CSE in smoking cessation counseling**	p value*	
		High (n = 64)	Low (n = 249)		
Sex				0.305	
  Male	205 (65.5)	45 (70.3)	160 (64.3)		
  Female	108 (34.5)	19 (29.7)	89 (35.7)		
Curricular phase				0.056	
  Pre-clerkship	172 (55.0)	27 (42.2)	145 (58.2)		
  Clerkship	141 (45.0)	37 (57.8)	104 (41.8)		
Smoking status				0.002	
  Non-smoker	251 (80.2)	41 (64.1)	207 (83.1)		
  Smoker	62 (19.8)	23 (35.9)	42 (16.9)	0.431	
Teaching methods				0.007	
  Lecture-based learning group	200 (63.9)	29 (45.3)	162 (65.1)		
  Blended learning group	113 (36.1)	35 (54.7)	87 (34.9)		
Scores of ≥ 60 on tobacco-related medical knowledge	39 (12.5)	15 (23.4)	24 (9.6)	0.003	
Notes.

Results are expressed as frequency (%).

* Chi-square test.

** Scores of 4 or higher being interpreted as high self-efficacy. CSE, counseling self-efficacy.

CSE in smoking cessation counseling

One-fifth (20.4%) of all students reported high self-efficacy for smoking cessation counseling, and the proportion of students with high self-efficacy was about 1.9 times higher (29.2%) in the blended learning group as compared to the lecture-based learning group (15.5%, p = 0.007, Table 2). Table 4 shows the characteristics of students according to self-efficacy. The percentage of students with high self-efficacy among smokers, those in the blended learning group, and those with tobacco-related medical knowledge scores ≥ 60 was significantly higher than those without (p = 0.002, 0.007, and 0.003, respectively). The proportion of tobacco-related medical knowledge scores ≥ 60 was only 9.6% among students who had low self-efficacy in smoking cessation counseling, while 23.4% students had a high self-efficacy, which was about 2.4 times the former (Table 4). Compared with low self-efficacy, the odds ratios (95% confidence interval) for those with high self-efficacy in smoking cessation counseling based on age, smoker, blended learning group, and with tobacco-related medical knowledge scores ≥ 60 were 1.1 (0.99–1.23), 2.73 (1.35–5.51), 2.32 (1.19–4.54), and 2.85 (1.23–6.63), respectively (Fig. 1).

Figure 1 Logistic regression plot of odds ratios and 95% CIs.

Discussion

More than 400,000 smokers registered in a smoking cessation support program by the National Health Insurance Service of Korea and received assistance from health care providers in 2017 (Oh, 2019). Although there are many reports that physicians’ advice for smoking cessation is effective for smokers in the health care setting, it is not known whether physicians or medical students are sufficiently trained in providing smoking cessation interventions. Therefore, this study examined items including the curricula of smoking cessation education, tobacco-related medical knowledge, and medical students’ CSE of smoking cessation treatment in four Korean medical schools. Thereafter, it evaluated whether current smoking cessation education was quantitatively or qualitatively appropriate for students, as well as analyzed factors affecting students’ CSE in smoking cessation treatment. As a result, in this study, students perceived that smoking prevalence was higher than the actual rate, tobacco-related mortality was lower than the actual rate, and smoking had less impact of smoking on cardiovascular disease than in reality. This overestimation of smoking prevalence and underestimation of health hazards were consistent with those observed in studies conducted by medical students in Italy (Grassi et al., 2012). It showed that students are not fully aware that smoking is a major global cause of disease and death.

We found that the knowledge about smoking cessation counseling techniques score (6.4%) was much lower than that of smoking cessation pharmacotherapy (22.7%) or tobacco-related illness (56.5%), as shown in Table 2. In addition, more than a quarter of the students said they did not remember having regular education in smoking cessation counseling at school. The lack of knowledge about practical counseling rather than knowledge of the disease was similar to the results of a survey conducted in the UK (Raupach et al., 2015). When physicians talked about smoking cessation to patients, focusing on the risk of smoking made the patients aware, but if the physicians were unable to advise the patients properly on how to quit smoking and guide them to stay as non-smokers, it would be difficult to say that the cessation treatment was truly a success (Raupach et al., 2015). From the results of this study, it seemed that students were not properly trained in the counseling skills most necessary in clinical settings. The effectiveness of traditional lecture-based education has been questioned for a long time, and many studies have found that alternative education, such as role-playing and standardized patient simulation, have been successful in improving attitudes toward tobacco use and smoking cessation counseling skills (Ockene et al., 2016; Park, Park & Hwang, 2019). In this study, students’ scores on partial or total tobacco-related medical knowledge showed significant differences according to the teaching method, and the differences were consistently observed regardless of gender, grade, and smoking status. Half of the medical schools included in this study only conducted lecture-type education. In the present study, only one-fifth (20.4%) of the students reported high self-efficacy for smoking cessation counseling due to education received from the medical school. In relation to education methods, about one-third of the students reported high self-efficacy in the blended learning group (15.3% in the lecture-based learning group), which was similar to the results of surveys conducted in Korea, Europe and Canada (Richmond et al., 2009; Kim, Issenberg & Roh, 2020; Bender et al., 2021). In contrast to the results of this study, even in a group that received classic lecture-based education in a randomized comparative study conducted in the United States (Caplan, Stout & Blumenthal, 2011), nearly half of the students showed high self-efficacy in smoking cessation counseling techniques and more than half in counseling of nicotine replacement therapy (Ockene et al., 2016). Asian students, including Korea, tend to underestimate self-efficacy beliefs compared to Western culture students due to the influence of East Asian cultures due to the pressure of success, fear of failure, and modesty etiquette (Putman, Wang & Ki, 2015). A previous comparative study on nutrition information also revealed differences in knowledge self-efficacy among college students between Korea and the US (Kim et al., 2020). While this might partly explain why students who showed high self-efficacy in this study was lower than expected, previous studies have analyzed that even though many more effective teaching methods have been developed, these have either not been utilized or education focused on the risks of smoking or simple medication rather than practical skills (Raupach et al., 2015). In addition, it can be assumed to have been caused by a difference in the expression of questions in the questionnaire or the cultural difference in self-expression. However, even within Korean culture, there are some differences between self-efficacy depending on the subject of study and circumstances (Jun, 2016; Lee & Young, 2018). In the present study, the average score of self-efficacy for smoking cessation counseling was 2.7, which was higher than the self-efficacy of physical activity (average score of 1.5) in Korean health college students (Lee & Young, 2018), and was lower than general self-efficacy (average score of 3.2) in Korean nursing college students (Jun, 2016).

In this study, a score of 60 or more on tobacco-related medical knowledge was correlated the most with high self-efficacy for smoking cessation counseling. There is controversy over the connection between CSE and training (Mehr, Ladany & Caskie, 2015; Mullen et al., 2015; Morrison & Lent, 2018), but this study confirmed that the teaching method may be also one of the ways to promote self-efficacy for smoking cessation counseling. Previous studies have shown that role-playing and modeling with visual images have been particularly helpful in improving CSE (Campbell et al., 2015; Botelho, Gao & Jagannathan, 2019). In our study, those with smoking experience tended to be more confident in smoking cessation counseling, as opposed to previous findings in which physicians who smoked thought that their smoking cessation recommendations would not help the patient (Huang et al., 2013; Reile & Parna, 2018). It is presumed that since the students in this study were relatively poorly trained in smoking cessation counseling skills at school, based on their experiences, smokers felt more confident in smoking cessation treatment. Medical schools A, B, C, and D reported having devoted 1, 4, 4, and 5 h, respectively, to smoking-related education. Except for school A, whose education hours were short, the remaining three schools’ education hours were comparable to those reported in a national survey of 22 medical schools in the UK (Raupach et al., 2015). Through the Korea Association of Medical Colleges, Korean medical schools set the same learning goals for smoking harm and cessation education. However, in view of the students’ overall levels of tobacco-related medical knowledge and self-efficacy for smoking cessation counseling being low, there is a need to reconsider the qualitative part of education, that is, its content and delivery methods.

In our study, the current smoking rate of medical students was 14.1% (male 19.8%, female 3.2%), which was lower compared to Korea’s present adult male smoking rate, but it was similar to the Korea’s present adult female smoking rate (Kim & Choe, 2019). By gender, approximately 40% of Korean adult males and 3% of Korean adult females were found to be smokers in 2016 (Chang et al., 2019). This can be interpreted as a decrease in the smoking rate of physicians leading to a decrease in the smoking rate of the public, as observed in most developed countries such as the US, Australia, and the UK (Cattaruzza & West, 2013). While most of the students smoked cigarettes, it is noteworthy that the types of cigarettes had diversified into electronic, smokeless, water, and hand-rolled cigarettes. In the absence of research focusing on the types of cigarettes used by Korean medical students, direct comparison was not possible. However, these medical students’ smoking behaviors were considered in the same context as the misconceptions of new cigarettes being safer, and the variety of cigarettes used by smokers due to the change in the Korean anti-smoking policy, which made it possible to target niche markets for new cigarettes, as well as easier to purchase products through the internet (Kim & Lee, 2017).

Conclusion

To summarize, we found that the blended learning method could lead to an increase in tobacco-related medical knowledge and CSE in smoking cessation. It is expected that with the implementation of the smoking cessation education curriculum in medical schools, interactive counseling techniques such as role-playing and standardized patient simulation, which are different from classical lecture-based education, will be introduced. This addition of more practical education to blended learning will increase students’ CSE for smoking cessation, and consequently, lead to high-quality smoking cessation treatment and reduction of smoking rates. However, the fact that only 20% of total students had high CSE for cessation counseling suggests that the Korean medical schools should convene to figure out what content needs to be taught and where to place it in the curriculum, although the high CSE rate was higher in the blended learning group. Therefore, it is necessary to be re-confirmed as a more well-designed randomized controlled study that reflects underlying factors that could also contribute to the changes in self-efficacy and knowledge. Additionally, based on this study, further research is also needed to develop educational methods that can improve CES or to verify the effectiveness of blended learning in various subjects of medical education.

Strength of the study

The present study focused on the overall areas necessary to assess whether Korean medical students were receiving appropriate smoking cessation education based on their curriculum, smoking status, experience of learning about tobacco in medical schools, tobacco-related medical knowledge, and CSE in smoking cessation. As a risk factor for bacterial and viral respiratory infections such as COVID-19 and MERD-CoV (Vardavas & Nikitara, 2020), the need for quitting smoking is increasingly emphasized, and smoking cessation education in medical schools has become a more important topic.

Limitations of the study

First, it was conducted in just four medical schools in Korea, the total response rate was not high, and the response rates for each grade were slightly different. The proportion of female students in this study is 34.5%, which is also similar to the proportion of female students (36.0%) in all Korean medical schools (Shin & Lee, 2020). A previous systematic review has shown that the response rate of online surveys was approximately 10% less than that of paper surveys (Fan & Yan, 2020). In addition, since this study was conducted with full voluntary participation of students, each school did not encourage or urge students to actively participate in the survey, which may be the reason for the rather low online reply rate. However, since the data that we analyzed was included in the 95% confidence interval with an error range of 5%, it seems that it was not insufficient to give statistical significance. Second, for this study we did not use a widely used questionnaire but specifically developed one based on our discussions and logical arguments. Since this study was not an attempt to objectively compare specific scores, the questionnaire also contained questions about the risks of smoking or smoking cessation counseling and its results were similar to previous studies on tobacco-related medical knowledge. We feel that the questions did not lower the quality of the study. Third, since this study was not designed as a randomized control study, there may be concerns about other confounding factors that undermine ability to interpret the data. For example, the grades in which the students were given smoking cessation education may have affected their knowledge scores, but comparisons were not possible since the schools’ curricula differed.

Supplemental Information

Supplemental Information 1 Data related to self-efficacy of smoking cessation counseling

Click here for additional data file.

Supplemental Information 2 Data related to self-efficacy of smoking cessation counseling (English version)

Click here for additional data file.

Supplemental Information 3 English-Korean codebook

Click here for additional data file.

Supplemental Information 4 Questionnaire (English)

Click here for additional data file.

Supplemental Information 5 Questionnaire (Korean)

Click here for additional data file.

Additional Information and Declarations

Competing Interests

Author Contributions

Ethics

Data Availability

The authors declare there are no competing interests.

Ara Cho and Sang Yeoup Lee conceived and designed the experiments, performed the experiments, analyzed the data, prepared figures and/or tables, authored or reviewed drafts of the paper, and approved the final draft.

Jeonggyu Lee, YunJin Kim and Byung Mann Cho analyzed the data, prepared figures and/or tables, authored or reviewed drafts of the paper, and approved the final draft.

Eunhee Kong, Minjeong Kim, Jinseung Kim, Dong Sik Jung and Seongho Han performed the experiments, prepared figures and/or tables, authored or reviewed drafts of the paper, and approved the final draft.

The following information was supplied relating to ethical approvals (i.e., approving body and any reference numbers):

The Institutional Review Board of Pusan National University Yangsan Hospital approved this research (IRB No. 05-2016-105).

The following information was supplied regarding data availability:

The raw measurements are available in the Supplementary Files.

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
