# Peer review of "Effects of education methods on self-efficacy of smoking cessation counseling among medical students"

_PeerJ, doi:10.7717/peerj.11408_

## Round 0.1 · original submission · Major Revisions

The authors should particularly address comments related to the experimental design. Any limitation on the data collection (interview, questionnaire, completion rate, sample size and demographics) and data analysis should be discussed and supported with literature.

In addition, as the reviewers point out, the manuscript literature review should be updated with recent literature and contradicting results addressed.

·

Basic reporting

Generally, the paper is clearly structured and well-written. Concerning literature references, it would be good to aim to incorporate more up-to-date literature, as quite some sources are 10+ years old already; also check small inconsistencies and mistakes in reporting references in-text. Further, the textual description of the results seems to be very complete but perhaps too elaborate, as this hampers its readability. Also, table 1 does not add information on top of what is mentioned in-text.

Experimental design

Several improvements should be made to the methods section of the paper, in order to make replicability of the study more likely.
- the informed consent procedure should be elaborated on;
- it would be important to elaborate on potential bias of asking coordinators to comment on their own education;
- the questions in the questionnaire require more explanation, e.g. answer scales/options, to understand what data was exactly collected;
- although a sample size estimation is given, this requires more elaboration.

Validity of the findings

Several improvements could be made to improve the validity of the study's findings, as described in the following suggestions.
- be mindful not to introduce (new) results (e.g. lines 282-287) in the discussion, but rather focus on an interpretation of results;
- the authors should be careful with making claims about causality (e.g. with lines 327-328 the authors hint at a potential causal relationship, which cannot be supported with the current findings);
- several limitations of the study are mentioned, but no supporting literature is provided to counter-argue these limitations;
- generally, more elaboration is warranted as to what the present study adds for the scientific community, e.g. describing its implications for research and practice.

Additional comments

Thanks for the opportunity to review this interesting paper. In addition to the feedback and suggestions already provided, I would advice the authors to also look at the following comments on the abstract and introduction of the paper.
Abstract:
- clear and concise abstract, though more information on data collection (e.g. paper-based vs digital) can be provided;
- information on data analysis is still lacking and should be added.

Introduction:
- important topics are described from broad to specific, though the authors could elaborate on the ‘new educational methods’ introduced in line 93.

·

Basic reporting

The article is good in language and professional design

Experimental design

Only 22.1% of the students completed the questinnaire and 19.8 % were smokers. This may not represent the real percentage of smokers. Because smokers are usually reluctant to repond voluntarily to such questionnaires. This should be included in the discussion.

Validity of the findings

Only 3.2 % of females in the study group were smokers which is very low.What is the percentage of female students in South Korean Medical Schools? Is this 34.5 % females close to the average percentge in medical schools?
Those with smoking experiences tended to be more confident in smoking cessation counseling which is contrary to previous findings in the literature (İcliet al1992,Brutons et al 2005).This may be related to inclusion of former smokers in the smoker group. They are usually more enthusiastic in smokinng cessation programs.

Additional comments

Congradulations for studying such an important public health problem

---

## Round 0.2 · accepted · Accept

However, before the manuscript publication steps start, please adjust your manuscript based on the comments found in the attached file.

·

Basic reporting

no comment

Experimental design

no comment

Validity of the findings

no comment

Additional comments

Thanks for giving me the opportunity to re-review this paper. The authors have adequately improved the paper based on the reviewers' feedback. Hence, I do not have any further comments. I wish the authors good luck in finalizing the paper.

·

Basic reporting

well done

Experimental design

No problem

Validity of the findings

Satisfactory additions were done related to my previous review

Additional comments

Congratulations for dealing with a very important subject of public health